



# Extreme drought event in AD 1637–1643 in North China: New insight from pollen records in Kaifeng City

Dexin Liu[1], Jianhua Ma[1, 2], Lei Gu[1], and Yanfang Chen[1]

[1] Institute of Natural Resources and Environment, Henan University, Kaifeng, China
5 [2] Collaborative Innovation Center on Yellow River Civilization of Henan Province, Kaifeng, China

*Correspondence to*: Jianhua Ma (mjh@henu.edu.cn)

**Abstract.** Long-lasting droughts usually result in water shortage and famine and even hinder the progress of human civilization. This paper presents the first study to identify the "Chong Zhen drought" event during the Late Ming Dynasty (AD 1637–1643) in a region with alluvial sediment. Using AMS[14]C dating, grain size and historical documents, we 10 determined the formation age of the sediment. Pollen records at 5–7.8 m, with the highest values for mesoxerophyte and xerophyte taxa such as Chenopodiaceae, *Nitraria* and *Ephedra* pollen, were found to provide evidence for the "Chong Zhen drought" event. Other pollen percentages were almost the lowest value of the entire core. However, sediment at 5–6.7 m also had a very high value of *Pinus*, which was mainly caused by fluvial transport and depositional processes. Chenopodiaceae, *Artemisia* and Asteraceae pollen are always transported and deposited together with coarser sediment during the flood 15 running stage, while pollen with thinner extine and air-sacs are always deposited in slow flow or hydrostatic conditions during the flood falling stage. In addition, pollen transported by wind from local and regional areas deposited on the ancient ground are always mixed with the sediment during the flood running stage with coarser until it covered by next flood. Our work helps shed light on the interpretation of the ancient vegetation and past climate based on pollen in alluvium, but it is important to make allowances for flood processes and sedimentary facies.

20 ## 1 Introduction

Understanding the process and mechanisms of climate change impacts and the resulting disaster events can enhance human response to global change (Costanza *et al.*, 2007). Numerous studies have found close and complicated correlations between climate change and historical social events, including, amongst others, population variation and migration, economic fluctuation, social harmony and crisis, and even dynastic transition (Haug *et al.*, 2003; Zhang *et al.*, 2008; Buckley *et al.*, 25 2010; Patterson *et al.*, 2010; Tol and Wagner, 2010; Büntgen *et al.*, 2011; Medina-Elizalde and Rohling, 2012; Pringle, 2012). In particular, drought is a recurring major natural hazard that has plagued civilizations through time and remains the "world's costliest natural disaster" (Herweijer *et al.*, 2006). When drought conditions persist for a number of years and spread across vast areas, these impacts become devastating.



A major long-lasting drought event that occurred in AD 1637−1643, known as the "Chong Zhen drought", was identified as the most severe and extreme drought event in North China in the past 2000 years (Ge, 2011). It covered almost 23 Chinese provinces, in which the most serious drought lasted for nine years (Ge, 2011). However, most researchers only focus on the context and consequence of the drought or its climate background from historical documents (Zhang, 2005; Peng and Xu, 2009; Ge, 2011), little is known about its stratigraphic evidence.

The past is the key to the present and future. During the past few decades, considerable attention has focused on paleoclimatology research due to its important role in the Past Global Changes (PAGES) project (Seppä and Bennett, 2003; Mackay, 2007; Seppä *et al.*, 2009; Xu *et al.*, 2010; Viau *et al.*, 2012; Evans *et al.*, 2013; Azuara *et al.*, 2015; Ladd *et al.*, 2015; Sun and Feng, 2015; Hao *et al.*, 2016; Flantua *et al.*, 2016). Vegetation dynamics are the most sensitive indicator of response to climate change. With their high production and fine preservation within sediments, pollen-based proxies have been preferred for reconstructing past climates and monitoring recent ones (Seppä and Bennett, 2003; Sun and Feng, 2015). Mensing *et al.* (2008) identified four periods of century-long droughts within the last 2000 years in the Great Basin by records of Chenopodiaceae and *Artemisia* pollen assemblage. Based on the records of *Nitraria* and *Ephedra* pollen assemblage and other proxy indexes, Yin *et al.* (2013) found three pronounced drought events during the mid-Holocene on the southeastern Inner Mongolian Plateau, China. These studies are mainly based on lacustrine sediments; however, very little is known about the relationship between alluvial pollen and drought events.

The flooding of the Yellow River has occurred many times near Kaifeng City, and these repeated floods have formed a relatively complete sedimentary sequence that provide a rare research opportunity for the exploration of stratigraphic evidence of the "Chong Zhen drought". In this paper, a 9-m long core was obtained from Jinming Campus of Henan University (JM core), which is located in the western suburbs of Kaifeng City, Henan Province, China. Here, based on the results of a detailed pollen analysis of JM core, we explore the response of regional vegetation to the "Chong Zhen drought" event. The purpose of this paper is to improve understanding of the facts and impacts of major abnormal climatic events during historical period, which will provide similar scenarios for human adaptation to future climate change and contribute to evaluating the possible influence of present or future extreme climatic events.

## 2   Materials and methods

### 2.1   Regional setting and sampling

Kaifeng City, one of the ancient capital cities of China, is located approximately twelve kilometers south of the Yellow River. It extends from 34°11'N to 35°01'N and 113°52'E to 115°15'E, with an average elevation of 72 m asl (Fig. 1). The climate type is the temperate continental monsoon climate with a mean annual temperature of 14℃ and a mean annual precipitation of 636 mm.



The natural potential vegetation is characterized by warm-temperate deciduous broad-leaved forest. At present, the original vegetation has mostly been replaced by farmland, orchards and plantations due to human activities. The grain crops are mainly dominated by *Triticum aestivum*, *Zea mays* and *Glycine max*. Economic crops mainly include *Arachis hypogaea*, *Citrullus lanatus* and *Gossypium hirsutum*. The dominant tree species are *Pyrus bretschneideri*, *Malus pumila*, *Paulownia elongata*, *Robinia pseudoacacia*, *Ulmus pumila*, *Salix matsudana* and *Populus tomentosa*. Herb species are dominated by *Eragrostis pilosa*, *Artemisia abrotanum*, *Salsola collina*, *Setaria viridis*, *Xanthium sibiricum* and *Bothriochloa ischaemum*.

According to historical records, the vicinity of Kaifeng was flooded many times. The earliest flood was recorded as early as 221 BC. Thereafter, floods were apparently not recorded for more than a thousand years until the beginning of the Song Dynasty, which was founded Kaifeng as its capital city. The Yellow River began to swing in the vicinity of Kaifeng in AD 1128, and floods became more prevalent. Six larger-scaled flood events happened respectively in the years of AD 1410, AD 1461, AD 1478, AD 1489, AD 1642 and AD 1841 inundated Kaifeng City during the Ming and Qing Dynasties, which not only damaged the city severely but also changed the surface sediment and natural landscape significantly (Cheng, 2003; Liu, 2009).

A 9-m-long sediment core (JM) was taken at 34°49′2.17″ N, 114°18′29.31″ E, elevation 74 m asl in April 2012 using a DPP-100 piston corer (9-cm in internal diameter). The sediment cores were transported back to the laboratory in sealed PVC pipes and described and subsampled shortly thereafter. A total of 78 samples for physico-chemical and palynological analysis were obtained at 10−20 cm intervals from the JM core.

## 2.2 Physico-chemical analysis

The grain size was measured at 10-cm intervals using a laser diffraction particle size analyzer (Mastersizer 3000, Malvern Co. Ltd., UK). Sample pretreatment included (1) adding $H_2O_2$ to remove organic matter and soluble salts, (2) using diluted 1 N HCl to remove carbonate and (3) using sodium hexametaphosphate to disperse aggregates. The wet color of the core was described using Munsell's soil color charts.

## 2.3 Pollen analysis

The 100−300 g samples (dry weight) were treated chemically according to the standard acetolysis procedure (Faegri and Iversen, 1989). Thirty-six percent HCl was used to dissolve calcareous minerals, 10% NaOH and 30% HF were used to remove the humic components and siliceous materials, respectively. Following the above, the pollen and spores were concentrated by heavy liquid ($ZnCl_2$, density 2.0) to separate them from undigested minerals. Finally, cellulose and humic debris were removed by acetolysis.

Pollen residues were stored in glycerol, mounted on microscope slides, and then scanned at 200× or 400× magnification with a Leica DM5500 microscope. Identification was aided by Wang *et al.* (1995), with comparison to the modern pollen reference collections. For each sample, a minimum of five slides were examined, and more than 400 terrestrial pollen grains were counted. Pollen percentages were calculated based on the sum of trees, shrubs and terrestrial herbs. Pollen diagram was



drawn using the TILIA software. Pollen zonation was performed using a stratigraphically constrained incremental sum of squares cluster analysis according to percentage data, facilitated by CONISS.

## 2.4 Radiocarbon dating

Radiocarbon dating took place using accelerator mass spectrometry (AMS) at the AMS Dating Laboratory at Peking University (Beijing, China). The $^{14}$C date was calibrated to calendar years before the present with the program CALIB Rev. 5.0.1 using the IntCal04 calibration data set (Reimer *et al.*, 2004). For this study, one sample (Lab number: BA130509) of plant macrofossil was selected and dated to $410 \pm 20$ BP, the calibrated age is AD $1460 \pm 30$ (7.85 m depth).

## 3  Results

### 3.1  Sedimentary cycle and chronology

Grain size is a major indicator used to divide sedimentary cycles. The sand-clay ratio of a core can reflect hydrodynamic changes. The greater the ratio, the stronger the hydrodynamics (Lecce and Pavlowsky, 2004). Therefore, sand-clay ratio can be used to conduct the division of sedimentary cycles. In this paper, wavelet periodic analysis of the sand-clay ratio was carried out with MATLAB 7.0 software (Fig. 2). As can be seen in Fig. 2, the core exhibit cycle changes from three types of scales (i.e., 1 m, 3 m and 6 m). Specifically, there are 1.5 sedimentary cycles in the 6 m scale, wherein the depth of 0–6 m is a complete cycle and 6–9 m for a half cycle (negative anomalies section). There are 2.5 cycles in the 3 m scale, wherein the depth of 1.8–4 m and 4–7.8 m are complete cycles and 7.8–9 m is a half cycle (negative anomalies section). There are more cycles in the 1 m scale, but only the depth of 0.5–1.8 m, 4–5.8 m and 5.8–7.5 m cycles are more stable. In addition, the other half of the cycle occurs at the depth of 0–0.5 m (positive anomalies section).

In March 2003, the Kaifeng Municipal Archaeological Team found a Song Dynasty cultural layer in the depth of 10–11.3 m in northeastern of the JM core (Liu, 2009). This archaeological discovery indicated that below the depth of 10 m was the ground surface of the Song Dynasty, and the 9-m-long core of this research was deposited after the Song Dynasty.

According to historical records (Cheng, 2003), the flooding of the Yellow River has affected Kaifeng City six times since the early 15th century, including floods occurred in AD 1410, AD 1461, AD 1478, AD 1489, AD 1642 and AD 1841 respectively. Additionally, the latest two  Yellow River floods in Kaifeng City occurred in AD 1642 and AD 1841 (Cheng, 2003; Liu, 2009), corresponding to the sediments below the surface of 4–7.8 m and 1.8–4 m sedimentary cycles, respectively (Fig. 2). Combined archaeological and dating data indicate that the 8 m depth, the grayblack paleosol, was the ancient ground surface for human activities after the Yellow River flood in AD 1489.

In the mid-20th century, the government introduced the Yellow River warping method for the purpose of controlling the hazards of sand storms around Kaifeng City. This formed a sludge layer of a certain thickness, which corresponded to the 0.5–1.8 m sedimentary cycle. The 0–0.5 m interval of the core is the modern artificial accumulation layer.



## 3.2 Comprehensive analysis of JM core

A comprehensive information schematic of the JM core is shown in Fig. 3. The sediment texture is composed mainly of silt loam, silt and sandy loam, and the sediment color is primarily brown and yellow-brown. The dark-brown and dark-gray colored layers, which mainly appeared in 2.5–2.7 m and 4.8–5.2 m depth of the core, indicate short-term hydrostatic reduction conditions. As seen from the vertical curve of sand-clay ratio (Fig. 3), the curve of cycle 2 fluctuated larger than cycle 1, probably due to temporal and spatial variation in the water flow during the flooding of the Yellow River in AD 1841 (Liu, 2009). However, an obvious secondary cycle is not found on the wavelet analysis chart because of the small sedimentary thickness. The flooding of the Yellow River in AD 1642 lasted three years, with widely varied hydrodynamic conditions, and only a mutation in 5.8 m depth was revealed on the wavelet analysis chart.

## 3.3 Palynological data

In total, 66 different types of terrestrial fossil pollen were identified from all samples in this study, and they include 23 tree taxa, 12 shrub taxa and 31 herb taxa (e.g., *Pinus*, *Picea*, *Quercus*, Moraceae, *Betula*, *Ulmus*, Elaeagnaceae, *Spiraea Salicifolia L.*, *Ostryopsis*, *Artemisia*, Chenopodiaceae, Cyperaceae, Gramineae, *Taraxacum*, Ranunculaceae, Asteraceae, Polygonaceae, etc.). In addition, one aquatic pollen, two algae and three fern spores (e.g., Typha; *Concentricystes*, Pediastraceae; *Selaginella sinensis*, Triletes and Monolete) were also identified.

The core can be divided into four pollen assemblage zones and four subzones according to the changes in the relative abundances of terrestrial pollen and the CONISS cluster analysis results (Fig. 4).

Pollen Zone 1 (7.8–9 m) corresponds to the sediment cycle $s_3$-h (Fig. 3), the pollen assemblage is dominated by herb component (82–96%), mainly including Chenopodiaceae (15–49%), Cyperaceae (10–35%), Gramineae (7–51%) and *Artemisia* (5–19%). *Pinus* is the main component of coniferous pollen, but with a low percentage value, lower than 6%. The broadleaved tree pollen (e.g., *Quercus* and *Betula*) percentages are also low.

Pollen Zone 2 (5–7.8 m) corresponds to the sediment cycle $s_3$-1, and the pollen assemblage is still dominated by herb component (38–97%, mean value 75%). It should be mentioned that mesoxerophytes taxa (e.g., Chenopodiaceae) percentages have the highest values of the entire core. This zone can be further divided into two subzones.

Pollen subzone 2a (6.7–7.8 m) corresponds approximately to the sediment cycle $s_1$-1 and is distinctively characterized by the highest percentage of herbs (78–97%, mean value 88%) of the entire core, mainly including Chenopodiaceae (21–77%, mean value 47%), *Artemisia* (4–39%), Cyperaceae (4–25%) and Gramineae (4–15%). The tree component abruptly decreased (mean value 7%) and mainly includes *Pinus* and *Quercus*. Pollen subzone 2b (5–6.7 m) corresponds roughly with the sediment cycle $s_1$-2. The assemblage is marked by increased arboreal pollen (up to 50%) at the expense of the herb pollen percentage (37–93%, mean value 64%). The tree types are mainly *Pinus* (1–27%), *Betula* (1–7%), *Quercus* (1–6%), *Picea* (0–7%), and Moraceae (0–7%), and *Pinus* is the major contributor to the increase in the tree pollen percentage.



Chenopodiaceae (6–70%, mean value 20%), *Artemisia* (1–13%) and Cyperaceae (1–19%) pollen percentages decreased, while Gramineae (7–36%) and Ranunculaceae (0–36%) pollen percentages increased in subzone 2b.

Pollen Zone 3 (2–5 m) corresponds roughly to the sediment cycle $s_3$-2. Compared with subzone 2b, this zone is distinctively characterized by dramatic decreases in the values of Chenopodiaceae and *Pinus* and also by dramatic increases in Ranunculaceae (1–43%, mean value 24%) and *Picea* (up to 10%) pollen percentages. Other pollen taxa, such as *Betula* and Moraceae, are slightly lower, while *Quercus*, Gramineae and Cyperaceae are slightly higher than that of subzone 2b. *Artemisia* percentages are almost unchanged from the previous subzone.

Pollen Zone 4 (0–2 m) corresponds roughly with the sediment cycles $s_1$-3 and $s_1$-h. Ranunculaceae, the dominant pollen type in the previous zone, decreased drastically and almost disappeared. Total herbaceous pollen percentages increased to approximately 81%, largely as a result of a marked increase of Gramineae (16–45%, mean value 32%) and Cyperaceae (10–42%, mean value 22%). Percentages of *Artemisia* (5–23%, mean value 11%) and Chenopodiaceae (5–12%, mean value 8%) also increased. Percentages of conifer pollen, dominated by *Pinus*, increased to 8% (4–15%), while *Picea* falls dramatically from a peak value of 10% to 1%. This pollen zone is also subdivided into two subzones. Compared with pollen subzone 4a, pollen subzone 4b has higher pollen percentages of *Pinus*, *Picea* and Cyperaceae but lower *Quercus*, Moraceae and Gramineae.

## 4   Discussion

### 4.1   Pollen response to the "Chong Zhen drought" event during the Late Ming Dynasty

The pollen assemblage of the JM core reveals an absolute dominance of herbaceous plant pollen (e.g., Chenopodiaceae, Gramineae, *Artemisia*, Cyperaceae and Ranunculaceae). Studies on alluvial pollen suggest the existence of hydrodynamics and taphonomic process would have influenced the deposition of pollen into the alluvial system and its preservation in the sedimentary record, as has been inferred to occur in alluvial sedimentary deposits from other regions (Hall, 1985; Fall, 1987; Xu *et al.*, 1996; Work *et al.*, 2005; Rojo *et al.*, 2012). According to these studies, pollen input and preservation in alluvial sedimentary archives could be controlled by pollen source, fluvial transport, depositional processes, sedimentary facies and post-depositional changes.

Chenopodiaceae and Asteraceae (including *Artemisia*) pollen, with relatively thicker pollen extine, are always transported and deposited together with coarse sand (Sangster and Dale, 1961; Hall, 1989). Thus, Chenopodiaceae pollen in the JM core was mainly derived from local areas and the middle and upper reaches of the Yellow River. Studies on modern pollen show that Chenopodiaceae pollen with over-represented and can be instructing regional ecology (Liu *et al.*, 1999; Li *et al.*, 2000; Cao *et al.*, 2010; Zhao and Li., 2013). Studies of relationships between modern pollen and vegetation show that the *Pinus* pollen is over-represented in the pollen record, and pine trees are locally present when the percentages are greater than 30% (Li *et al.*, 2000; Cao *et al.*, 2010; Zhang *et al.*, 2014). In the JM core, abundant *Pinus* pollen occurs in subzone 2b, with the




highest value reaching 27%, suggesting that *Pinus* pollen was mostly transported by floods from the middle and upper reaches of the Yellow River.

Therefore, the vertical variation of pollen assemblage in JM core primarily reflects the different periods of vegetation changes in the areas near the middle and upper reaches of the Yellow River during the last 500 years. The pollen of the JM core is composed of local pollen, regional pollen and extra-regional pollen while regional pollen is the most essential component. Local pollen, which is deposited mainly during the intermittent period, is a reflection of local vegetation.

Human impacts on the natural vegetation, especially on arboreal taxa, were widespread and intensifying during the last 500 years (Li *et al.*, 2006). As a result, the herb pollen assemblage in the JM core can better reflect dry and wet changes. Pollen zone 1 (7.8–9 m), which was mainly deposited in approximately AD 1460, has a high value of Chenopodiaceae pollen percentage (mean value 35%), suggesting an obvious drought climate during the Middle Ming Dynasty period. Numerous studies show that the AD 1440s may be the beginning of a drought climate during the Ming Dynasty, and the drought degree in AD 1450–1490 is next to the Late Ming Dynasty (Yang *et al.*, 2014; Zheng *et al.*, 2014; Lee *et al.*, 2015; Hao *et al.*, 2016). This is consistent with the pollen record in this study.

In pollen zone 2 (5–7.8 m), the highest values of mesoxerophyte taxa such as Chenopodiaceae, *Taraxacum* and Asteraceae pollen percentage are found, accompanied by the appearance of xerophyte taxa such as *Nitraria* and *Ephedra* pollen (Fig. 5). However, other pollen percentages are almost the lowest value of JM core (Fig. 4). Sediments of this interval are associated with the the Yellow River flood in AD 1642, and its high content of xerophyte and mesoxerophyte taxa correspond well with the "Chong Zhen drought" event during the late Ming Dynasty.

### 4.2 Reasons for the differences between pollen subzone 2a and 2b

Different from pollen subzone 2a (6.7–7.8 m), subzone 2b (5–6.7 m) is distinctively characterized by a high percentage of tree pollen, especially *Pinus* pollen, reaching the highest value of the entire core (Fig. 4). The reasons for this phenomenon may be the following two aspects. (1) The flood depositions of the Yellow River mainly brought herb pollen from the middle reaches and floodplains. During the "Chong Zhen drought" period, the decline in vegetation coverage in the middle reaches of the Yellow River may have led to an enhancement of wind erosion, largely as a result of a marked increase of surface pollen migration by the wind. In addition, the precipitation variability tends to increase when the climate is relatively dry, and more pollen were moved by water erosion from surface soil to the river. During a flood of the Yellow River there may be two flood sedimentary layers: one was deposited during flood running period, and another was deposited during flood falling stage. During the flood running stage, as the river water has higher flow velocity, only the pollen with thicker extine and greater gravity (e.g., Chenopodiaceae, *Artemisia* and Asteraceae) can be deposited with coarser sediment, however, during the flood falling stage, the river flow rate declined obviously, so the pollen with thinner extine (e.g., Moraceae) and air-sacs (e.g., *Pinus*) can be deposited with fine sediment (usually with clay) (Sangster and Dale, 1961; Traverse and Ginsburg, 1966; Hall, 1989; Xu *et al.*, 1996; Brown *et al.*, 2007; Albert, 2014). Pollen subzone 2b is precisely in the negative anomalies section of the cycle, also indicating the power of the flooding water diminished and even became hydrostatic





deposition (Fig. 4). (2) As the fine sediment (clay) was deposited during the flood falling stage, it will be the new ground surface after the flood. The pollen and sedimentary materials from local and regional areas transported by wind can be deposited on the ground until it was covered by next flood. In particular, *Pinus* pollen can be easily transported long distances by wind due to its air-sacs. Additionally, dry air caused by intense wind facilitates anther cracking and pollen dispersion (Shang *et al.*, 2009). So, the pollen from local and regional areas transported by wind usually mixed with the sediment during the flood running stage with coarser sediment.

## 5 Conclusions

From our study of AMS $^{14}$C dating, historical records and wavelet analysis of sand-clay ratio in the JM core, we found that the sediment in the 0.5–1.8 m, 1.8–4 m and 4–7.8 m intervals in the core were formed by irrigation in the mid-20th century, flooding of the Yellow River in AD 1841 and AD 1642, respectively. Pollen records in the 5–7.8 m interval provide evidence for the "Chong Zhen drought" event, wherein mesoxerophyte taxa such as Chenopodiaceae, *Taraxacum* and Asteraceae in the 6.7–7.8 m interval reached the highest values, accompanied by the appearance of xerophyte taxa such as *Nitraria* and *Ephedra* pollen. Other pollen percentages are almost the lowest value of the entire core. However, sediment in the 5–6.7 m interval also has a very high value of *Pinus*, which was mainly caused by fluvial transport and depositional processes. Chenopodiaceae, *Artemisia* and Asteraceae pollen are always transported and deposited together with coarser sediment during the flood running stage, while pollen with thinner extine and air-sacs are always deposited in slow flow or hydrostatic conditions during the flood falling stage. Then, the surface sediment developed into ancient ground. Pollen transported by wind from local and regional areas deposited on the ground are always mixed with the sediment during the flood running stage with coarser until it covered by next flood.

## Acknowledgements

This work was funded by National Natural Science Foundation of China (No. 41171409), Humanities and Social Science Projects by Ministry of Education of China (No. 12JJD790023), and Program for Innovative Research Team (in Science and Technology) in University of Henan Province (No. 16IRTSTHN012).

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





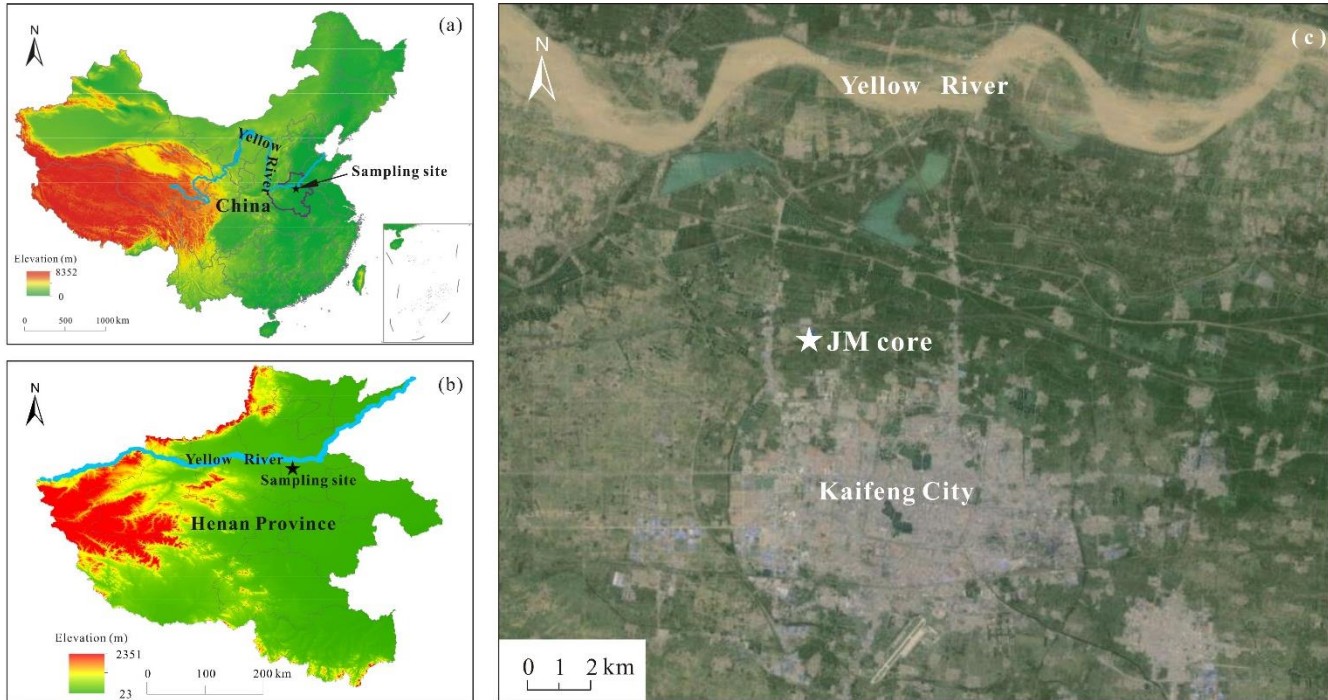

**Figure 1. (a)** Elevation map of the geographical location of the sampling site in China and **(b)** Henan province; **(c)** Map showing the location of the JM core.



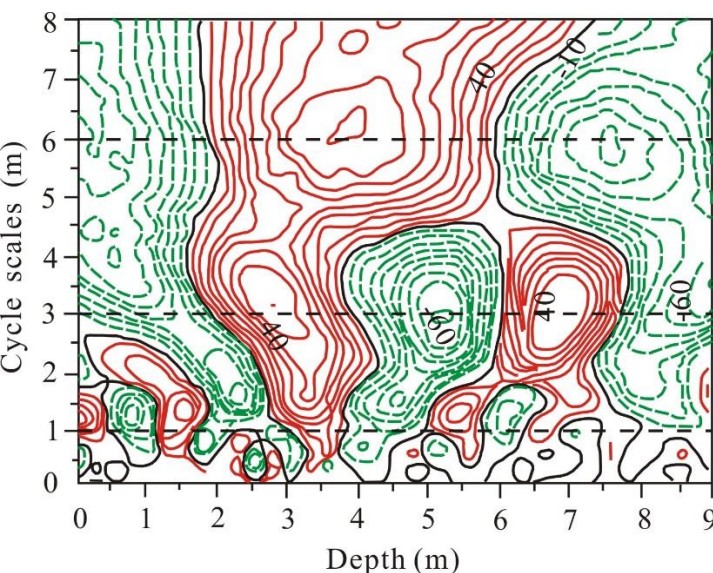

**Figure 2.** Contour map of the sand-clay ratio by using wavelet analysis of the JM core.







**Figure 3.** Comprehensive information schematic of the JM core. In writing the notation of a wet color, the order is hue, value, and chrome.

For instance, a color of 10YR in hue, 4/ in value, and /3 in chrome, the notation is 10YR 4/3.




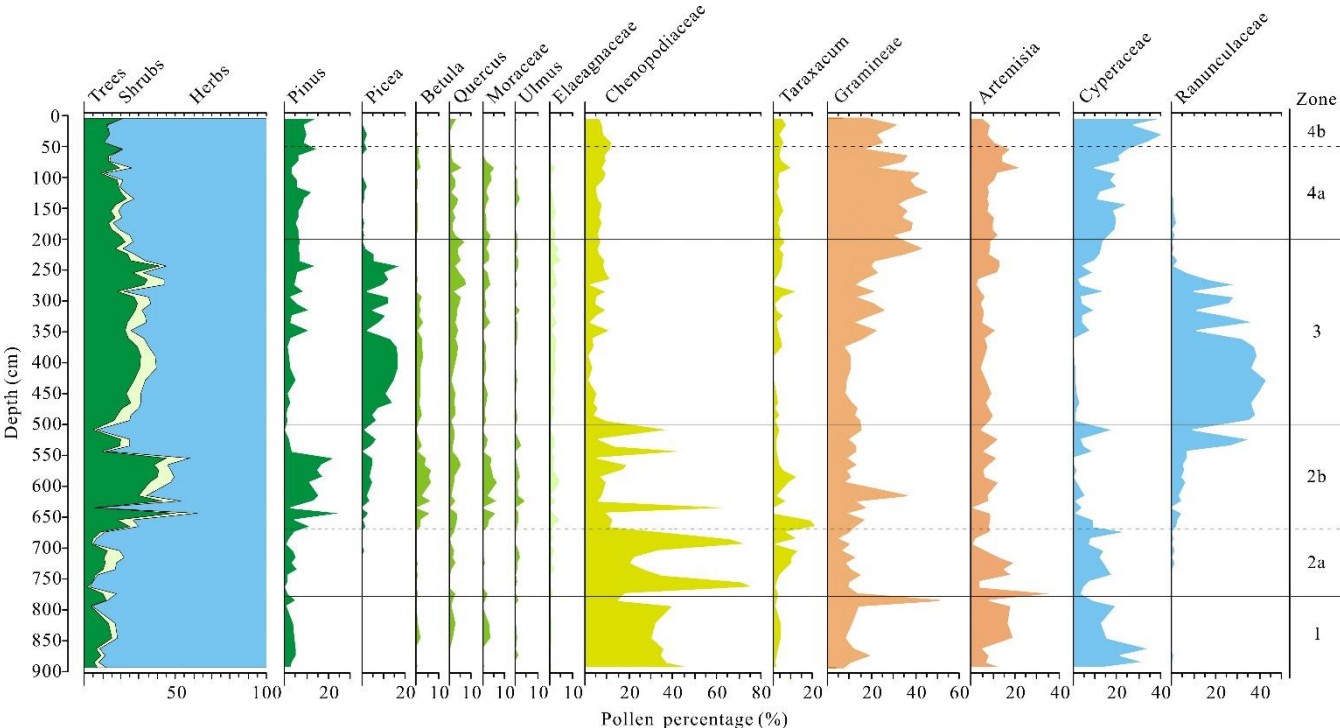

**Figure 4.** Pollen percentage diagram of the JM core.



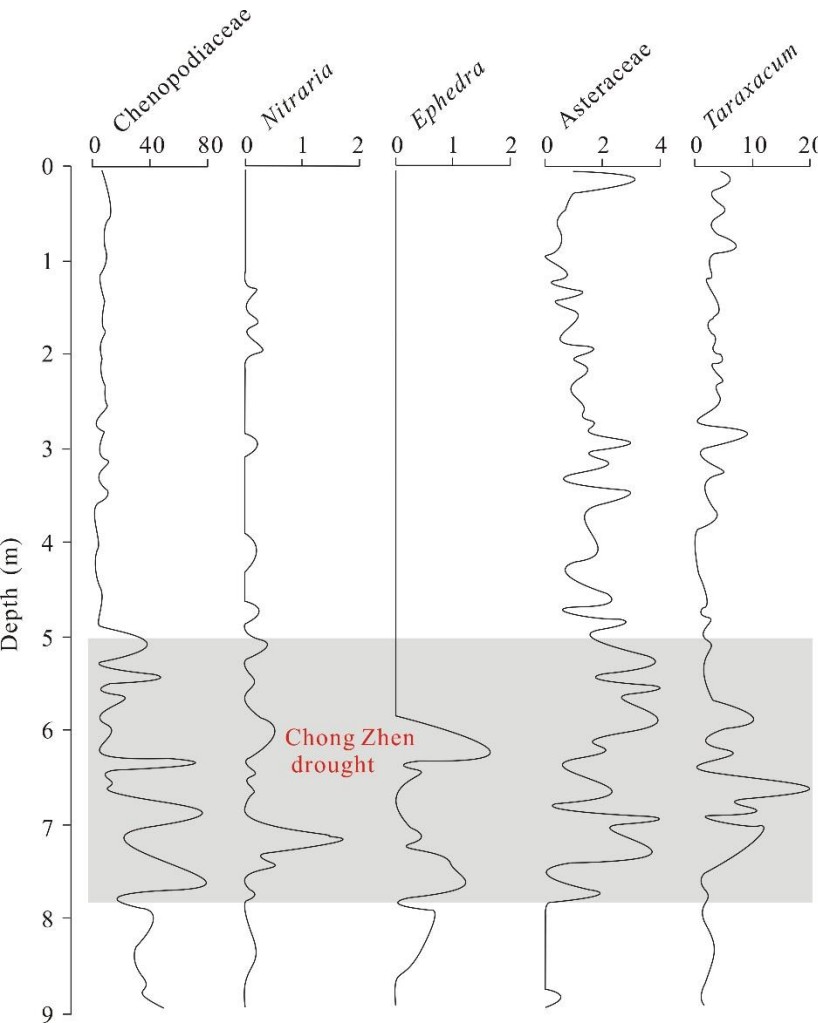

**Figure 5.** Pollen percentage of drought-tolerant taxa in the JM core.