# Peer review of "Extreme drought event in AD 1637–1643 in North China: New insight from pollen records in Kaifeng City"

_Climate of the Past, 2016_

## Referee Comment (RC1) · Anonymous Referee #1 · 4 Jan 2017

The paper presented by Liu et al. aims at the identification of the "Chong Zhen drought" event in a fluvial section near Kaifeng City, China. Although the topic of extreme climate events (i.e. droughts) and its influence on the society is of high scientific interest, the manuscript does not significantly contribute to this discussion. The manuscript lacks a clear focus and has little further implications. In this review, I will focus on the stratigraphic and sedimentological aspects of the manuscripts as these issues are within my expertise.

General comments:

1 Fluvial setting is not convincing, missing age control

The authors have chosen a fluvial setting in order to characterize a relatively short-lived drought event (AD 1637-1649). I would be very skeptical to extract detailed information

from such a fluvial setting as it is by nature very fragmentary and in no way continuous. Therefore, it is – in my view - almost impossible to attribute a certain depth section to this drought phase of 6 years. And if it was a drought phase, what kind of sediment would have been deposited during this climate period? I would expect little to no deposition or even erosion during dry conditions. This is also the case for the time period between the flood events of AD 1642 and AD 1841, where no sediment was deposited. For these reasons, this fluvial setting is not a suitable archive for a detailed paleoenvironmental reconstruction.

2. Grain size wavelet analysis

The authors measured the grain size of the deposits and the first thing they present is a wavelet analysis of the grain size data (i.e. Fig. 2). I would suggest that the authors first present the grain size data and then do further analysis on the grain size data. However, I highly question the use of wavelet analysis in a depth scale. Wavelet analysis gives information about potential underlying periodicities and this most useful in a time scale. Why should there be any cycles in a depth scale. It would imply that all flood events are equal-thick and the deposits have a similar grain size evolution. That is not what I would expect. Also in the text, the authors mention the 3m cycles, but the exact length of the two cycles is either 2.2m or 3.8m (see Page 4, Line 16). So this is not very convincing for a stable 3m cycle. Furthermore, what would be the interpretation of the 1m cycle (Page 4, Line 17)? Instead of the wavelet analysis, the study needs a detailed sedimentological description together with the grain size analysis showing the grading of the different intervals that can be related to flood or flood pulses.

Specific comments:

Page 4, Line 6: Use the up-to-date version of CALIB 7 and IntCal13

Figure 5: The grey marked interval is NOT the Chong Zhen drought according to Fig. 3 (here this depth interval is labeled as the Yellow River flood in AD 1642)!!

For all these reasons, I do not recommend the manuscript at this stage.

---

## Author Comment (AC1) · 10 Jan 2017

Dear referee,

We are truly grateful to your critical comments and thoughtful suggestions on our manuscript (Extreme drought event in AD 1637–1643 in North China: New insight from pollen records in Kaifeng City. No. cp-2016-122). Based on these comments and suggestions, careful modifications will be made in the revised manuscript. Below you will find our point-by-point responses to your comments.

**Replies to the Referee:**

The "Chong Zhen drought" event was one of the most severe and extreme drought event in North China in the past 2000 years. However, most researchers only focus on the context and consequence of the drought or its climate background from historical documents, little is known about its stratigraphic evidence. The purpose of this paper is to explore the response of regional vegetation to the "Chong Zhen drought" event by a fluvial setting near Kaifeng City (see Page 2, Lines 1-5; 20-24).

1. **The reviewer's comment:** The authors have chosen a fluvial setting in order to characterize a relatively short-lived drought event (AD 1637-1649). I would be very skeptical to extract detailed information from such a fluvial setting as it is by nature very fragmentary and in no way continuous. Therefore, it is – in my view - almost impossible to attribute a certain depth section to this drought phase of 6 years. And if it was a drought phase, what kind of sediment would have been deposited during this climate period? I would expect little to no deposition or even erosion during dry conditions. This is also the case for the time period between the flood events of AD 1642 and AD 1841, where no sediment was deposited. For these reasons, this fluvial setting is not a suitable archive for a detailed paleoenvironmental reconstruction.

**The authors' answer:** In the alluvial sequence, frequent flooding can lead to a high accumulation rate of alluvial deposits and this sedimentation is considered as "continuous". Conversely, lower rate of alluvial deposition can cause a longer time exposure to the surface and deposits are greatly influenced by the pedogenesis. At this point sedimentation is considered as "intermittent". The alternation and combination of "continuous" and "intermittent" constitute a relatively complete record of depositional environment ([1] Behrensmeyer, A. K., Willis, B. J., and Quade, J.: Floodplains and paleosols of Pakistan Neogene and Wyoming Paleogene deposits: a comparative study, Palaeogeogr. Palaeoclimatol. Palaeoecol., 115(4), 37–60, doi: 10.1016/0031-0182(94)00106-I, 1995. [2] Bridge, J. S., Willis, B. J., and Behrensmeyer, A. K.: Architecture of Miocene overbank deposits in northern Pakistan; discussion and reply, J. Sediment. Res., 65(3b), 401–407, doi: 10.1306/D4268261-2B26-11D7-8648000102C1865D, 1995. [3] Kraus, M. J.: Lower Eocene alluvial paleosols: Pedogenic development, stratigraphic relationships, and paleosol/landscape associations, Palaeogeogr. Palaeoclimatol. Palaeoecol., 129(3–4), 387–406, doi:10.1016/S0031-0182(96)00056-9, 1997. [4] Wang, H. Y., Shi, Y. C., Yu, P. T., Wang, M. H., Hao, J. M., and Li, L.: Alluviums of the early and middle Holocene in the Quzhou area, the southern Hebei plain and palaeoenvironment inferences. Quat. Sci., 22(4), 381–393, doi: 10.3321/j.issn:1001-7410. 2002. 04. 011, 2002.).

During the drought periods, the decline of the vegetation coverage in surroundings of Kaifeng city and its upper reaches may have led to an enhancement of wind erosion, which is largely resulted by a marked increase of surface pollen migration by the wind. In addition, the precipitation variability

tends to be increasing when the climate is relatively dry, and more pollen were moved from surface soil to the river by water erosion. During a flood of the Yellow River, pollen in the flood sediments of the Yellow River mainly derived from the surface runoff, indicating there is a better overall appearance of the basin vegetation. Therefore, alluvial pollen assemblage in the flood period makes better explanation to the regional paleoenvironment ([1] Hall, S. A.: Pollen analysis and paleoecology of alluvium, Quat. Res., 31(3), 435–438, doi: 10.1016/0033-5894(89)90052-5, 1989. [2] Peng, W., Lin, X., Huang, X., Liu, S., and Jia, W.: Preliminary study on assemblages characteristics of waterborne pollen and accumulated pollen from the Yellow River at Lanzhou station, J. Palaeogeogr., 18(4), 691–698, doi: 10.7605/gdlxb.2016.04.051, 2016. [3] Xu, Q., Yang, X., Chen, W., Meng, L., and Wang, Z.: Alluvial pollen on the North China plain, Quat. Res., 46, 270–280, doi: 10.1006/qres.1996.0066, 1996. [4] Xu, Q., Yang, X., and Yang, Z.: Relationship between pollen assemblages and vegetation in alluvial sediments of Luanhe River Basin. J. Palaeogeogr., 6(1), 69–77, doi: 10.3969/j.issn.1671-1505.2004.01.008, 2004.).

In this paper, less chronological data were acquired. But we can precisely determine the formation age of the sediment based on AMS$^{14}$C dating, historical documents and the archaeological discovery (see Page 3, Lines 9-12; Page 4, Lines 4-7; 19-30). We will improve this part in the revised version.

2. **The reviewer's comment:** The authors measured the grain size of the deposits and the first thing they present is a wavelet analysis of the grain size data (i.e. Fig. 2). I would suggest that the authors first present the grain size data and then do further analysis on the grain size data. However, I highly question the use of wavelet analysis in a depth scale. Wavelet analysis gives information about potential underlying periodicities and this most useful in a time scale. Why should there be any cycles in a depth scale. It would imply that all flood events are equal-thick and the deposits have a similar grain size evolution. That is not what I would expect. Also in the text, the authors mention the 3m cycles, but the exact length of the two cycles is either 2.2m or 3.8m (see Page 4, Line 16). So this is not very convincing for a stable 3m cycle. Furthermore, what would be the interpretation of the 1m cycle (Page 4, Line 17)? Instead of the wavelet analysis, the study needs a detailed sedimentological description together with the grain size analysis showing the grading of the different intervals that can be related to flood or flood pulses.

**The authors' answer:** Thanks for your comment. Wavelet analysis is available in this research. Sedimentary strata was the product of continuous deposition during a long period. They embodied rich environmental information about the sedimentary events and their profile properties varied from time to time, so that the extracted information along the vertical profile can be taken as another form of time information. Therefore, space-frequency analysis, which is transformed by time-frequency analysis of wavelet, was used to reveal the sequence characteristics and sedimentary cycles of strata in different temporal and spatial scales. With the development of sequence stratigraphy, geologists have applied wavelet analysis to sedimentary cycle division of well logging curve since 1980s ([1] Goldhammer, R. K., Dunn, P. A., and Hardie, L. A.: Depositional cycles, composite sea-level changes, cycle stacking patterns, and the hierarchy of stratigraphic forcing: examples from Alpine Triassic platform carbonates, Geol. Soc. Am. Bull., 102(5), 535-562, doi: 10.1130/0016-7606(1990)102<0535:DCCSLC>2.3.CO;2, 1990. [2] Goldhammer, R. K., Lehmann, P. J., and Dunn, P. A.: The origin of high-frequency platform carbonate cycles and third-order sequences (Lower Ordovician El Paso Gp, west Texas): constraints from outcrop data and stratigraphic modeling, J. Sediment. Petrol., 63(3), 318-359, doi: 10.1306/D4267AFA-2B26-11D7-

8648000102 C1865D, 1993. [3] Deng, H.: Discussion on problems of applying high resolution sequence stratigraphy, J. Palaeogeogr., 11(5), 471–480, doi: 10.7605/gdlxb.2009.05.001, 2009. [4] Wang, G., Xu, J., Yang, N., Lai, J., and Zhao, X.: Using wavelet frequency analysis to divide sedimentary sequence cycles and isochronous correlation, Geol. J. Chin. Univ., 19(1), 70–77, 2013. [5] Zhao W, Qiu L, Jiang Z, and Chen Y.: Application of wavelet analysis in high-resolution sequence unit division, J. Chin. Univ. Petrol. (Ed. Nat. Sci.), 33(2), 18–22, doi: 10.3321/j.issn:1673-5005.2009.02.004, 2009.).

In general, the grain size of sediments are expressed by sand-clay ratio, which is a traditional and available indicator of sedimentary cycle division. Besides, sedimentary cycles divided by wavelet analysis are more accurate and less subjective than those were divided by visual judgment. As the expert expected, Wavelet variance map of the sand-clay ratio of the JM core was provided to present the accurate scales (Fig. a). We will improve this chapter in the revised version.

[Figure]

**Figure a**. Wavelet variance map of the sand-clay ratio of the JM core

**Specific comments:**

1. **The reviewer's comment:** Page 4, Line 6: Use the up-to-date version of CALIB 7 and IntCal13

**The authors' answer:** As suggested by the reviewer, we will replace the CALIB Rev. 5.0.1 and IntCal04 by OxCal v4.2.3 (Bronk Ramsey, C.: OxCal 4.2.2 Manual, https://c14.arch.ox.ac.uk, 2013.) and IntCal13 (Reimer, P. J., Bard, E., Bayliss, A., …, and van der Plicht, J.: IntCal13 and Marine13 radiocarbon age calibration curves 0-50,000 years cal BP. Radiocarbon, 55(4), 1869–1887, doi: 10.2458/azu_js_rc.55.16947, 2013.) respectively in the revised version.

2. **The reviewer's comment:** Figure 5: The grey marked interval is NOT the Chong Zhen drought according to Fig.3 (here this depth interval is labeled as the Yellow River flood in AD 1642)!!

**The authors' answer:** Sediments of this interval (5–7.8 m) are associated with the Yellow River flood in AD 1642. During the flood, both sediment and topsoil pollen from the Middle Reaches of the Yellow River are deposited in downstream, which can reflect the overall appearance of the basin vegetation. Therefore, sediments brought by the flood can indicate the drought event. Please see the detailed answer above.

3. **The reviewer's comment:** For all these reasons, I do not recommend the manuscript at this stage.

**The authors' answer:** Based on the arguments that raised above, we believe that the revised version will merit publishing in Climate of the Past Journal.

We appreciate your effort, detailed comments, and useful suggestions. We will take up your constructive comments to improve our manuscript.

Sincerely yours,
Dexin Liu, Jianhua Ma, Lei Gu & Yanfang Chen
10 January, 2017

---

## Referee Comment (RC2) · Anonymous Referee #2 · 11 Jan 2017

Comments to the Author:

I found the topic and general approach to be quite interesting and of potential interest to the journal. Although the "Chong Zhen drought event" has been recorded in historical document, it has not been confirmed by geological evidence. Based on sedimentary record and pollen analyses from a sediment core near the Kaifeng City, the authors found that Pollen records in the 5‒7.8 m interval provide evidence for the "Chong Zhen drought" event, which mesoxerophyte pollen taxa such as Chenopodiaceae, Taraxacum and Asteraceae reached the highest values, accompanied by the appearance of xerophyte pollen taxa such as Nitraria and Ephedra. However, there are still some problems that need to be improve. The manuscript should be published once my comments have been addressed.

The detailed problems are as follows:

Page 3 Line 8: "...the vicinity of Kaifeng used to be flooded many times." Line 10: "...when it became its capital city."

Page 4 Line 19: "In March 2003, the Kaifeng Municipal Archaeological Team......" You referred to a Song Dynasty archaeological layer but did not give the age or clearly state whether it was found at this site or elsewhere. In order to sustain your argument and convince your readers, your explanation need to describe clearly. Line 23: "...the Yellow River flood have disturbed Kaifeng City..."

Page 6 Lines 19‒21: "Studies on alluvial pollen suggest...in alluvial sedimentary deposits from other regions." The meaning is unclear. The existence of hydrodynamics and taphonomic process are mainly influencing the pollen deposition and pollen preservation. Line 25: "Chenopodiaceae and Asteraceae (including Artemisia) pollen, with relatively thicker pollen extine", in addition to thicker pollen extine, it may also has the higher gravity.

Page 7 Line 6: The "intermittent period" are also deposited, just not by flood deposition. It may be better replaced by "there was no flood deposition in study area". Line 18: "...with the Yellow River flood..., and its higher pollen content of xerophyte and mesoxerophyte plants..."

————————————————

---

## Short Comment (SC1) · 2 Feb 2017

In this paper, the authors reported a new insight from pollen records to identify the extreme drought event in AD 1637‒1643 in North China. This is very interesting research since the studies during the past few decades were mainly based on lacustrine sediments, however, the relationship between alluvial pollen and drought events have not been explored yet. The results provide a wide view to identify the "Chong Zhen drought" event during the Late Ming Dynasty (AD 1637‒1643) in a region with alluvial sediment, and I therefore suggest the acceptance of this paper for publication in "Climate of the Past" after minor revisions.Wavelet analysis is an effective tool to analyze the periodical change of time series data. In this paper, the wavelet analysis of sediment grain size in the depth scale was carried out. In my opinion, the conclusion obtained from wavelet analysis in the time scale instead of in the depth scale

is more reliable, also can clarify visually the periodic change of drought-flood evolution. Furthermore, I would suggest that the author add the wavelet variance analysis to determine accurately the periodicities and their intensity.

---

## Author Comment (AC2) · 10 Feb 2017

The comment was uploaded in the form of a supplement:
http://www.clim-past-discuss.net/cp-2016-122/cp-2016-122-AC2-supplement.pdf

---

## Author Comment (AC3) · 10 Feb 2017

We are truly grateful to Dr. Chu Chunjie for his interest in this manuscript and for sharing the critical comments and thoughtful suggestions. In the following, we sketch how we plan to address the two issues brought out by Dr. Chu Chunjie in the revision.

1. Wavelet analysis is an effective tool to analyze the periodical change of time series data. Space-frequency analysis, which is transformed by time-frequency analysis of wavelet, was used to reveal the sequence characteristics and sedimentary cycles of strata in different temporal and spatial scales. With the development of sequence stratigraphy, geologists have applied wavelet analysis to sedimentary cycle division of well logging curve since 1980s (Goldhammer et al., 1990, 1993; Deng, 2009; Zhao et al., 2009; Wang et al., 2013). As Dr. Chu suggested, the conclusion obtained from

wavelet analysis in the time scale instead of in the depth scale is more reliable. However, sedimentary strata embodied rich environmental information about the sedimentary events and their profile properties varied from time to time, so that the extracted information along the vertical profile can be taken as another form of time information. Therefore, wavelet analysis of sediment grain size in the depth scale was carried out in this paper.

Main references as follows: Deng, H.: Discussion on problems of applying high resolution sequence stratigraphy, J. Palaeogeogr., 11(5), 471‒480, doi: 10.7605/gdlxb.2009.05.001, 2009. Goldhammer, R. K., Dunn, P. A., and Hardie, L. A.: Depositional cycles, composite sea-level changes, cycle stacking patterns, and the hierarchy of stratigraphic forcing: examples from Alpine Triassic platform carbonates, Geol. Soc. Am. Bull., 102(5), 535-562, doi: 10.1130/0016-7606(1990)102<0535:DCCSLC>2.3.CO;2, 1990. Goldhammer, R. K., Lehmann, P. J., and Dunn, P. A.: The origin of high-frequency platform carbonate cycles and third-order sequences (Lower Ordovician El Paso Gp, west Texas): constraints from outcrop data and stratigraphic modeling, J. Sediment. Petrol., 63(3), 318-359, doi: 10.1306/D4267AFA-2B26-11D7-8648000102 C1865D, 1993. Wang, G., Xu, J., Yang, N., Lai, J., and Zhao, X.: Using wavelet frequency analysis to divide sedimentary sequence cycles and isochronous correlation, Geol. J. Chin. Univ., 19(1), 70‒77, 2013. Zhao W, Qiu L, Jiang Z, and Chen Y.: Application of wavelet analysis in high-resolution sequence unit division, J. Chin. Univ. Petrol. (Ed. Nat. Sci.), 33(2), 18‒22, doi: 10.3321/j.issn:1673-5005.2009.02.004, 2009.

2. As suggested by Dr. Chu Chunjie, wavelet variance analysis of the sand-clay ratio of the JM core will be provided to present the accurate scales in the revised manuscript.

---

## Author Comment (AC4) · 10 Feb 2017

Dear referee,

We are truly grateful to your critical comments and thoughtful suggestions on our manuscript (Extreme drought event in AD 1637–1643 in North China: New insight from pollen records in Kaifeng City. No. cp-2016-122). Based on these comments and suggestions, careful modifications will be made in the revised manuscript. Below you will find our point-by-point responses to your comments.

1. The reviewer's comment: Page 3 Line 8: "...the vicinity of Kaifeng used to be flooded many times." Line 10: "...when it became its capital city."

The authors' answer: As suggested by the reviewer, we will rewritten the sentence in
the revised version.

2. The reviewer's comment: Page 4 Line 19: "In March 2003, the Kaifeng Municipal Archaeological Team. . . . . . ." You referred to a Song Dynasty archaeological layer but did not give the age or clearly state whether it was found at this site or elsewhere. In order to sustain your argument and convince your readers, your explanation need to describe clearly. Line 23: ". . .the Yellow River flood have disturbed Kaifeng City. . ."

The authors' answer: Thanks for your comment. In March 2003, the Kaifeng Municipal Archaeological Team found a Song Dynasty cultural layer (AD 960–1127) in the depth of 10‒11.3 m, located in northeast of the JM core about 200 m. We will add this to the revised version. As suggested by the reviewer, we will replace "the flooding of the Yellow River has affected Kaifeng City" by "the Yellow River flood have disturbed Kaifeng City".

3. The reviewer's comment: Page 6 Lines 19‒21: "Studies on alluvial pollen suggest. . .in alluvial sedimentary deposits from other regions." The meaning is unclear. The existence of hydrodynamics and taphonomic process are mainly influencing the pollen deposition and pollen preservation. Line 25: "Chenopodiaceae and Asteraceae (including Artemisia) pollen, with relatively thicker pollen extine", in addition to thicker pollen extine, it may also has the higher gravity.

The authors' answer: Thanks for your comment. Studies on alluvial pollen suggested that the existence of hydrodynamics and taphonomic process would have influenced the pollen deposition and its pollen preservation in the alluvial sediment. We will rewritten the sentence to make the statement clearer. As suggested by the reviewer, we will add "higher gravity" to the revised version.

4. The reviewer's comment: Page 7 Line 6: The "intermittent period" are also deposited, just not by flood deposition. It may be better replaced by "there was no flood deposition in study area". Line 18: ". . .with the Yellow River flood. . ., and its higher pollen content of xerophyte and mesoxerophyte plants. . ."

The authors' answer: As suggested by the reviewer, we will rewritten the two sentences in the revised version.

We appreciate your effort, detailed comments, and useful suggestions. We will take up your constructive comments to improve our manuscript.